# DeepDebug: Fixing Python Bugs Using Stack Traces, Backtranslation, and Code Skeletons

## Abstract

The joint task of bug localization and program repair is an integral part of the software development process. In this work we present `DeepDebug`, an approach to automated debugging using large, pretrained transformers. We begin by training a bug-creation model on reversed commit data for the purpose of generating synthetic bugs. We apply these synthetic bugs toward two ends. First, we directly train a backtranslation model on all functions from 200K repositories. Next, we focus on 10K repositories for which we can execute tests, and create buggy versions of all functions in those repositories that are covered by passing tests. This provides us with rich debugging information such as stack traces and print statements, which we use to finetune our model which was pretrained on raw source code. Finally, we strengthen all our models by expanding the context window beyond the buggy function itself, and adding a skeleton consisting of that function's parent class, imports, signatures, docstrings, and method bodies, in order of priority. On the QuixBugs benchmark, we increase the total number of fixes found by over 50%, while also decreasing the false positive rate from 35% to 5% and decreasing the timeout from six hours to one minute. On our own benchmark of executable tests, our model fixes 68% of all bugs on its first attempt without using traces, and after adding traces it fixes 75% on first attempt.

## 1 Introduction

The dominant paradigm in automated program repair is the generate-and-validate approach, which our work follows. In this setting we assume the existence of a suite of test functions that identify the existence of a bug. We must then localize the bug and consider candidate fixes until finding a patch that satisfies the test suite.

Throughout our experiments, we work with synthetic bugs in which the error has already been localized to a single buggy method. We take as input the buggy function, along with additional context depending on the experiment, such as surrounding context from the function's file and a stack trace that exposes the buggy function. We feed that input to our sequence-to-sequence transformer, which attempts to generate the fixed function in its entirety.

In our deployment scenario, we also attempt to localize the bug using the stack trace. At present we apply a simple heuristic based on which lines of the stack trace come from the developer's own code, considering the most recently called lines to be the most suspect. In future we are interested in improving our heuristic using an encoder transformer that reranks methods given the stack trace.

## 2 Related Work

### 2.1 Fault Localization

Curiously, the standard approach to bug localization eschews stack traces and instead uses spectrum-based fault-localization (SBFL). In this approach, lines of code are ranked in order of suspiciousness based on how many failing tests execute them. One example is the DStar formula (Wong et al., 2014). Given a statement $s$ that is executed by $failed(s)$ failing tests and $passed(s)$ passing tests, it

Figure 1: Our training pipeline. Following best practice we take advantage of extensively pretrained transformers, in this case using the same `DeepDev-py` sequence-to-sequence model used to finetune PyMT5 (Clement et al., 2020). We first use commit data to train both a baseline bugpatcher model along with a bug-creator model. Our bug-creator models feeds twenty times as much data to `DeepDebug` (backtrans). Finally we finetune this model on neural bugs injected into functions that have executable tests and produce traces, thus obtaining its final evolution `DeepDebug` (traces).

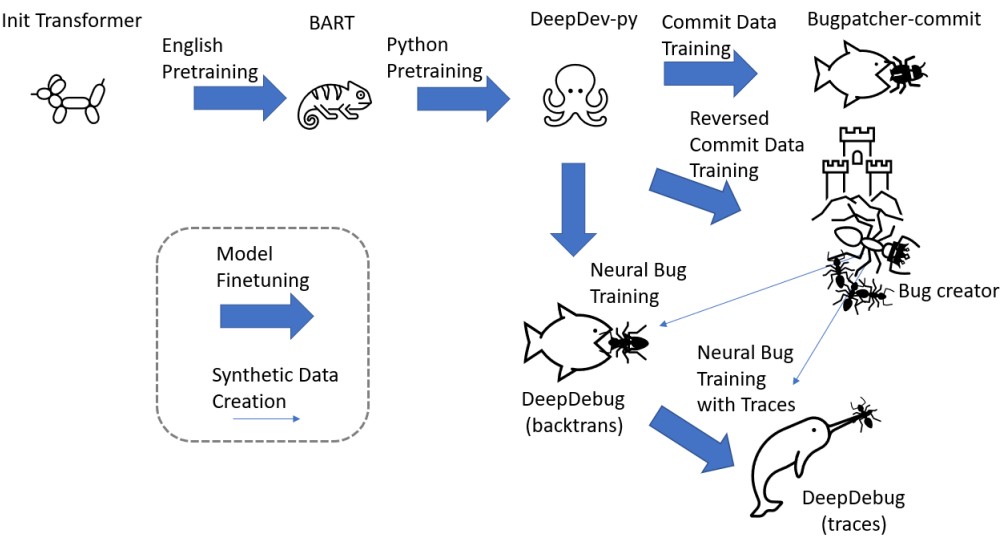

computes the suspiciousness score

$$S(s) = \frac{failed(s)^e}{passed(s) + (totalfailed - failed(s))}$$

where $e$ is an exponent like 2.

A notable exception to the SBFL approach to fault localization is the task of build repair, in which the localization information comes from the compiler message rather than a test suite. Gao et al. use crash traces to query Stack Overflow and produce repairing edits (Gao et al., 2015). The `DeepDelta` model was trained on build data from 300 million lines of Java code from Google projects (Mesbah et al., 2019). Using a Neural Machine Translation approach, `DeepDelta` fixes half of all errors resulting from mismatched method signatures or a missing symbol on its first attempt.

## 2.2 FALSE POSITIVES

A pervasive problem in the field is the fact that tests are incomplete oracles. Many tools produce patches that are merely test-suite adequate more often than they are genuine fixes. For instance, when evaluated on the QuixBugs challenge (Lin et al., 2017), a benchmark consisting of one-line bugs in classic algorithms, a set of ten bugpatching tools found solely false positives for nine functions, while only actually finding any genuine fix for seven (Ye et al., 2019). Only three models found more genuine fixes than false positives, in all cases exactly one more genuine fix. The neural approach CoCoNuT does somewhat better, finding thirteen genuine fixes along with seven false positives (Lutellier et al., 2020). A survey found similarly disappointing results on the larger benchmark Defects4j (Just et al., 2014), a collection of real-world Java bugs (Durieux et al., 2019). In stark contrast, we observe an excellent false positive rate with our approach.

One can also evaluate bug-patching models without executing tests. The *Patches in the Wild* benchmark frames bug-patching as a sequence-to-sequence problem as in our approach, which enables the use of NLP metrics (Tufano et al., 2019). The authors consider an edit to be a bonafide fix only if it is exactly the same as the edit the original developer made when committing it. SequenceR (Chen et al., 2018) both narrows and expands on the *Patches in the Wild* dataset by focusing on one-line changes only, while also providing additional context beyond the buggy method by including other methods' signatures, similarly to our use of code skeletons. This extended context gives a 15% relative boost.

## 2.3 SYNTHETIC DATA

For certain bug-types it is possible to generate millions of synthetic bugs to train on. Devlin et al. (Devlin et al., 2017) train an RNN on Python to fix incorrect comparison operators, the mistaken use of "is" vs. "is not", variable misuse, and forgotten "self" accessors. Overall, they achieve 86% accuracy on synthetic bugs and 41% on real-life bugs. Kanade et al. (Kanade et al., 2019) pretrain a BERT-style model "CuBERT" on a larger Python dataset and then finetune on a related suite of synthetic bugs, achieving over 90% accuracy.

Backtranslation is a more generic form of data augmentation, in which a model is trained to translate target data to source data (Edunov et al., 2018). In our case, we have an abundance of bug-free data mined from GitHub. We train a backtranslation model to create synthetic bugs and augment the training data for our goal task of fixing bugs. Backtranslation is also put to use for the related task of grammatical error correction (Kiyono et al., 2019).

## 2.4 PRETRAINING

Various task-agnostic pretraining approaches like BERT, BART, T5, and GPT-3 (Devlin et al., 2017; Lewis et al., 2020; Raffel et al., 2019; Brown et al., 2020) have seen large performance gains on a diverse array of benchmarks. These models are typically pretrained using a denoising objective on a large corpus of text that has been synthetically corrupted using bidirectional or causal masking, as well as more arbitrary noising such as randomly deleting, replacing, or permuting source tokens. Pretraining has also shown large improvements for the task of program repair (Drain et al., 2021; Lu et al., 2021).

## 3 MODEL

We reuse the 406M-parameter sequence-to-sequence transformer with twelve encoder layers and twelve decoder layers which was pretrained as described under section 4.1.

When experimenting with stack traces, we allot 1024 tokens for the code skeleton, and up to 896 tokens for the trace. In order to accommodate this larger context we thus need to expand the transformer's positional embedding matrix. To this end we use axial embeddings, as inspired by reformer (Kitaev et al., 2020). In our implementation, we duplicate the first 896 of the pre-existing 1024 positional embeddings, generate a random axial vector, and add that vector to each of the 896 duplicate embeddings. This approach outperformed randomly initialized embeddings in preliminary experiments.

## 4 DATA

We work with four different training datasets: Raw python code used for pretraining, commit data used for training a neural bug-creator and bug-patcher, methods extracted from the raw code in which we insert both neural bugs so as to train an even stronger bug-patcher, and, finally, methods that pass executable tests. For this last dataset, we also obtain the list of lines executed by each test, and we obtain another bug-patching dataset by again inserting synthetic bugs and rerunning the passing tests, which allows us to finetune a bug-patcher on stack traces, error messages, and print statements. We also experiment with giving our bug-patcher models either only the focal buggy method or also a 'skeleton' of the entire file that prioritizes data such as function signatures.

## 4.1 PRETRAINING

We build on the DeepDev transformer platform. We reuse the 406M-parameter DeepDev Python transformer that was warmstarted from FaceBook's BART model and then further pretrained using a spanmasking objective (Lewis et al., 2020; Clement et al., 2020). The pretraining data consists of 200,000 public Python repos filtered to have at least five stars. Pretraining took place on a DGX-2 box for three weeks. Note that the DeepDev tokenizer has appended whitespace tokens, such as the four-space and eight-space tokens, which boosts throughput and the effective context length.

To minimize any risk of leakage, we consistently restrict to the same validation and test repositories, in particular to those repositories used in CodeSearchNet (Husain et al., 2019).

## 4.2 COMMIT DATA

We traverse the commit history of 100,000 Python repositories that were filtered to have at least ten stars. We further filter to all commits whose message contains the word "fix", roughly one fifth of all commits. Based upon inspecting many examples, it seems that this simple filter is approximately as precise as more restrictive filters that insist on phrases like "patch bug" or "fix error". Nevertheless, the data is still extremely noisy.

This commit data serves two purposes for us. First, it allows us to train an edit model which is biased towards constructive, bug-fixing edits. We can evaluate such a model directly on bug-fixing, or finetune it on more filtered bug data. Second, we can reverse the input and output and train an edit model biased towards destructive, bug-inducing edits. We can use this model to create neural bugs to greatly augment our training data. This backtranslation approach has already proven useful elsewhere in nlp.

Since we are interested in fixing buggy methods, we consider each method edited by each commit. As we are only interested in nontrivial edits, we normalize each method before and after the commit and discard the edits that do not affect the normalized code. To normalize, we strip comments, replace string and numeric literals with placeholders like 'STR_LIT', and standardize whitespace. Finally we are left with 1.1 million, nontrivially edited methods.

We experiment with giving solely the focal method and its edit as input and output, as well as putting more context into the input, as described in the section on Code Skeletons.

Table 1: Cross-entropy loss for training two transformers, one on commit data and the other on reversed commits. The two models are evaluated on both forward and backward edits, and with and without code skeletons. The cross-entropy losses are five times better than those normally reported for generating Python code since the editing task is relatively easy ((Svyatkovskiy et al., 2020; Clement et al., 2020)). Furthermore, the losses for reversed edits are a third lower than for forward edits. The forward model is 6% better at forward edits than the reverse model, and the reverse model is in turn 6% better at reverse edits. Both models perform 2% better with skeletons than with only the focal method. We report the results in base two on the test set.

|  | Forward Edit Model | Backward Edit Model |
|---|---|---|
| Forward Edits With Skeletons | .240 | .258 |
| Forward Edits, focal method only | .248 | .265 |
| Reverse Edits, With Skeletons | .163 | .155 |
| Reverse Edits, focal method only | .167 | .157 |

## 4.3 SYNTHETIC BUGS

### 4.3.1 NEURAL BUGS

We use the term 'neural bugs' to refer to synthetic bugs that are created using a neural edit model, such as the one we trained to revert bug-fixing commits. Using neural bugs for data augmentation has many appealing features. Flexible neural models can generate near arbitrary edits that are drawn from the distribution of mistakes developers actually make. For instance, a neural edit model might swap get_key with get_value, whereas a simple heuristic approach might make unlikely or random swaps, such as switching get_key with reverse_list. Furthermore, this approach is nearly language agnostic, as we can reuse our framework for mining commits, and only need a parser to extract classes and methods along with the constituent parts need to form the code skeletons.

We present three examples of neural bugs. In our first example, the bug-creator model replaces kwargs.pop with kwargs.get, as used elsewhere in the function. In the second example, our bug-creator model replaces a check for a bucket name starting with self.name with a stricter equality check. In the third example, our model deletes a break statement.

| Fixed | Buggy |
|-------|-------|
| ```python
def get_key(self, *args, **kwargs):
    """Pass 'force' to _get_key_internal()
    in the headers
    since the call signature of _get_key_internal
    can not be changed.
    """
    if kwargs.pop('force', None):
        headers = kwargs.get('headers', {})
        headers['force'] = True
        kwargs['headers'] = headers

    return super(Bucket, self).get_key(
                        *args, **kwargs)
``` | ```python
def get_key(self, *args, **kwargs):
    """Pass 'force' to _get_key_internal()
    in the headers
    since the call signature of _get_key_internal
    can not be changed.
    """
    if kwargs.get('force', None):
        headers = kwargs.get('headers', {})
        headers['force'] = True
        kwargs['headers'] = headers

    return super(Bucket, self).get_key(
                        *args, **kwargs)
``` |

Figure 2: Our bug-creator model replaces kwargs.pop with kwargs.get, as used elsewhere in the function.

| Fixed | Buggy |
|-------|-------|
| ```python
def tearDown(self):
    for bucket in self.boto_conn.get_all_buckets():
        if bucket.name.startswith(self.name):
            for key in bucket.list():
                key.delete()

            bucket.delete()

    for key in self.redis.keys(
            tpl.connection + '*'):
        self.redis.delete(key)
``` | ```python
def tearDown(self):
    for bucket in self.boto_conn.get_all_buckets():
        if bucket.name == self.name:
            for key in bucket.list():
                key.delete()

            bucket.delete()

    for k in self.redis.keys(
            tpl.connection + '*'):
        self.redis.delete(k)
``` |

Figure 3: Our bug-creator model replaces a check for the bucket name starting with self.name with a stricter equality check.

We also observe our model injecting errors such as replacing a dot accessor with a bracket accessor, replacing comparison operators like ">=" with "<", truncating chained function calls, deleting return lines, wrapping return values in objects such as tuples and dictionaries and conversely forgetting to wrap objects, replacing precise errors like IndexError with different errors like ValueError, intuitively misnaming variables such as self.result instead of self._result, and mistakenly copying by reference instead of copying by value. In particular we observe all heuristic bugs that have previously been reported in the literature.

Nevertheless, we note a few shortcomings of our approach in generating neural bugs. Our models often hallucinate member variables that do not exist, although this is partially remedied by using a longer context than the focal method alone. This is a minor issue, although using a longer context is especially important when training on bugpatching, as bugs are often otherwise underspecified. Our approach over-represents bugs that arise from partial refactoring, such as changing member variable names or function signatures. Our approach under-represents bugs that are caught before being committed, such as syntactic errors and typos. Our approach is also prone to generating superficial changes, such as trivial changes to formatting, local variable names, and practically equivalent edits such as replacing an "is not None" check with a "!= None" check.

One glaring shortcoming of our model, although not necessarily our approach, is that our model prematurely generates the end-of-sentence token in around a fifth of all instances during random sampling. Indeed, these cases represent a large majority of all edits containing syntax errors. This occurs most often during difficult edits, such as when inside a nested scope such as f(["..., especially when manipulating strings or numbers or when invoking user-defined functions or objects from user-defined classes. We could artificially decrease the probability of generating the end-of-sentence token, although we do not explore this option and instead filter out 90% of bugs that contain syntactic errors after they are produced.

### 4.3.2 HEURISTIC BUGS

We use the term 'heuristic bugs' to refer to synthetic bugs that are manually created using simple rules, such as deleting a line or swapping two parameters in a function call.

| Fixed | Buggy |
|---|---|

```python
def _generate_SAX_single(self,
              sections, section_height, value):
  sax = 0
  for s in sections:
    if value >= sections[s]:
      sax = s
    else:
      break
  return sax
```

```python
def _generate_SAX_single(self,
              sections, section_height, value):
  sax = 0
  for s in sections:
    if value >= sections[s]:
      Sax = s
  return sax
```

Figure 4: Our bug-creator model deletes the break, leading sax to take the last small value in sections, instead of the first. Our model also introduces a trivial indentation error on the penultimate line.

Previous work has typically considered five or fewer types of bugs at a time, such as replacing a binary operator (e.g. using != instead of ==), using the wrong variable, forgetting the 'self.' accessor, or deleting code.

### 4.3.3 RAW METHODS

Since we are interested in data augmentation via synthetic bugs, we can make use of the plenitude of relatively bug-free code on GitHub. In doing so we potentially expand our dataset of target methods over twenty-fold compared to working solely with functions mined from bug-fixing commits. Furthermore, we can scale up nearly arbitrarily by creating multiple buggy version of each method. For this paper we restrict to 1.3M functions from 10K repositories, near parity with the commit data, and scale to 18M bug-fix pairs via backtranslation. We leave further scaling to future work.

### 4.4 METHODS WITH EXECUTABLE TESTS

There is a world of opportunity for debugging code that can actually be executed, especially if there is an accompanying test to verify the execution was correct. A typical debugging session involves homing in on the suspect block of code with the help of the stack trace, inserting print statements and breakpoints in an approximate binary search, modifying and executing snippets of code to gauge one's intuition, searching StackOverflow for interpretations of error messages and examples of API usage, talking to ducks, and incrementally making progress. In contrast, the baseline neural model is impoverished, and can only do the equivalent of staring at a piece of code for a few seconds before decisively writing the correction one token at a time.

At a minimum, the generate-and-validate approach enabled by executable tests allows for improved performance via multiple chances. For instance, in the domain of short Java methods, researchers have consistently seen top-20 accuracy triple that of the top-1 accuracy (Allamanis et al., 2017; Tufano et al., 2019). Indeed, truly random edits can be sure of passing a test suite with sufficiently many tries, although prior work has shown that these edits can aggressively overfit, as tests found in the wild are not perfect oracles and can often be gamed by simply deleting functionality (Qi et al., 2015).

### 4.4.1 TRACES

In addition to using tests to triage inaccurate edits, we incorporate information from tests into training in three different ways: appending the error message to the buggy method, additionally appending the stack trace, and further providing the values of all local variables at the point of failure, the last of which is conveniently provided by our testing framework Pytest .

### 4.4.2 COLLECTING PASSING TESTS

To collect executable tests at training scale we start with the 200K repositories used for pretraining and filter to 35K repositories containing tests and a setup.py or requirements.txt file. For each of these repositories we execute Pytest in a unique container, ultimately collecting passing tests from 10K repositories.

**Example Pytest Trace**

```
self = HobokenApplication(name='TestCatchallFilters',
...debug=False)
match = re.compile(b'.*')
func = <function TestCatchallFilters.after_setup.
...<locals>.after_all at 0x7f5f70a81d30>

    def add_after_filter(self, match, func):
>       filter_tuple = self.__build_filter(match, func)
E       AttributeError: 'HobokenApplication' object
...has no attribute '_HobokenBaseApplication__build_filter'

func        = <function TestCatchallFilters.after_setup.
...<locals>.after_all at 0x7f5f70a81d30>
match       = re.compile(b'.*')
self        = HobokenApplication(name='TestCatchallFilters',
...debug=False)

hoboken/application.py:480: AttributeError
```

Figure 5: Example output for a single frame of the stack trace produced by Pytest . In general there are several frames for a given stack trace. From top to bottom we can see: the input variables (self, match, and func in this case); the head of the function up until the final line executed, denoted with a '>' sign; the error message; the values of all local variables; and finally the file path, line number, and error name. Line breaks and ellipses have been manually added for readability. We append the Pytest trace to the buggy code skeleton, and experiment with withholding the heads and local variables.

### 4.4.3 TESTING SYNTHETIC BUGS

After filtering to functions with passing executable tests and inserting neural bugs, we then rerun the tests to collect Pytest traces and filter out edited functions that still pass their tests and thus are not actually buggy.

### 4.5 CODE SKELETONS

Much previous work on bug-fixing has used a very abbreviated context consisting solely of the buggy focal method. However, many bugs are not detectable or remediable without a larger context. In particular, we often need to reference user-defined functions, classes, and global variables, as well as the list of imported libraries.

To that end, we experiment with fleshing out the entire 1024-token window using the highest-priority lines within the focal file. We place highest priority on the focal buggy method, followed by its class's definition if applicable, imports, its class docstring and attributes, globals, other methods' signatures, other methods' docstrings, and finally global variables and other methods' bodies. We also single out the focal method using the special book-ending comments "# target edit" and "# end".

## 5 EXPERIMENTS AND RESULTS

We experiment with training on backtranslation data, adding skeletons, and adding the Pytest stack trace.

### 5.1 BACKTRANSLATION

In our first experiment we compare between training only on forward commit data vs. training on synthetic bugs produced via backtranslation. We evaluate using cross-entropy on the holdout data.

Evaluated on holdout forward commit data as shown in table 2, `DeepDebug` (backtrans) attains up to 10% better loss. Surprisingly, the backtranslation model actually performs worse on reversed commit data, which is consistent with its bias towards constructively fixing buggy functions.

Overall `DeepDebug` is much stronger than the previous state of the art. In table 3 we report our model's results on the QuixBugs challenge, a benchmark of forty classic algorithms with small synthetic bugs, with nearly equivalent versions in both Python and Java (Lin et al., 2017). The original QuixBugs challenge was for developers to fix as many bugs as possible given one minute apiece.

We limit our model to generating one hundred patches via random sampling, which is approximately the number that can be generated and evaluated over the span of one minute. We improve upon

Example Skeleton

```python
import psycopg2
import psycopg2.extras
from dpostools.schemes import schemes
from dpostools.utils import dictionify
from dpostools import api, exceptions

class DbConnection:
    def __init__(self, user, password, host='localhost',
            database='ark_mainnet'):
    def connection(self):

class DbCursor:
    def __init__(self, user, password, host='localhost',
            database='ark_mainnet', dbconnection=None):
    def description(self):
    def execute(self, qry, *args, cur_type=None):
    def fetchall(self, cur_type=None):
    def fetchone(self, cur_type=None):
    def execute_and_fetchall(self, qry, *args, cur_type=None):

    # target edit
    def execute_and_fetchone(self, qry, *args, cur_type=None):
        self.execute(qry, *args, current_type=cur_type)
        return self.fetchtwo(current_type=cur_type)

    # end

class DposNode:
    def __init__(self, user, password, host='localhost',
            database='ark_mainnet', ):
    def account_details(self, address):
    def node_height_details(self):
    def check_node_height(self, max_difference):
    def all_delegates(self):
    def current_delegates(self):
    def payouts_to_address(self, address):
    def transactions_from_address(self, address):
    def all_votes_by_address(self
```

Figure 6: Example skeleton using just 400 tokens. Note the inclusion of the entire buggy focal function (with its name highlighted for the reader) including the control tokens '# target edit' and '# end', as well as the imported functions. Due to limited space, the skeleton can only fit all signatures from the focal class, along with several more signatures from outside the focal class. The skeletons we use in practice contain more tokens and thus tend to be fleshier i.e. contain more docstrings and function bodies.

Table 2: Cross-entropy results for training two transformers, one trained on commit data and the other on patching neural bugs. As in table 1, the two models are evaluated on commit data. Note that the neural bugpatcher does comparatively worse on reversed edits vs. forward edits. The neural bugpatcher actually does worse when using skeletons on commit data, presumably due to the fact that commits typically edit multiple functions. Finally, the neural bugpatcher outperforms the forward edit model when evaluated on individually edited functions.

|  | Forward Edit Model | DeepDebug (backtrans) |
|---|---|---|
| Forward Edits With Skeletons | .240 | .243 |
| Forward Edits, focal method only | .248 | .221 |
| Reverse Edits, With Skeletons | .163 | .196 |
| Reverse Edits, focal method only | .167 | .174 |

Table 3: Results on the QuixBugs Benchmark. We use a timeout of one minute for each bug and perform joint bug-localization and program repair. In contrast, prior works use a timeout of several hours and often assume that the buggy line has already been localized.

|  | Suite of Ten non-neural Models (Durieux et al., 2019) | CoCoNut (Lutellier et al., 2020) | DeepDebug (backtrans) (ours) |
|---|---|---|---|
| Number of Bugs Fixed | 7/40 (18%) | 13/40 (33%) | 21/40 (53%) |
| Number of False Positives | 9 | 7 | 1 |
| True Positive Rate | 44% | 65% | 95% |
| Number of Verbatim Fixes | – | – | 13 |

the previous state of the art for number of bugs found by over 50% (Lutellier et al., 2020), while simultaneously decreasing the false positive rate from 35% to 5%. Notably all our models generate many duplicate edits, due to the low-perplexity of this task, which suggests room for improved sampling. Note that these results are even more impressive given the hints and timeouts provided to previous models. For instance, CoCoNuT is explicitly told which line contains the bug and allows six hours to find a patch (Lutellier et al., 2020). Similarly, five of the non-neural tools collectively found 122 test-suite adequate patches for the longest increasing subsequence algorithm (Durieux et al., 2019), suggesting thousands of attempts.

## 5.2 SKELETONS

In the second experiment we compare between training and evaluating using only the focal function as input vs. giving the entire skeleton as input. The neural bugpatcher sees a large 25% loss decrease when using skeletons when evaluated on neural bugs, as shown in table 4. Surprisingly, the neural bugpatcher actually does worse when using skeletons on commit data as reported in table 2, presumably due to the fact that commits typically edit multiple functions.

Table 4: The neural bugpatcher sees a large 25% cross-entropy loss decrease when using skeletons when evaluated on our large test set of neural bugs. These neural bugs are in general not executable.

|  | `DeepDebug` (backtrans) |
| --- | --- |
| Neural Bugpatching, With Skeletons | .306 |
| Neural Bugpatching, focal method only | .409 |

## 5.3 TRACES

In the third experiment we append the Pytest stack trace to the buggy input. We expand the context window using axial embeddings in order to fit these additional tokens.

We filter to a benchmark of 523 neural bugs from 100 repositories in our validation set. We observe impressive performance, and, contrarily to our cross-entropy results, a large boost from using traces. The top-10 patch success rate for `DeepDebug` (backtrans) is 90%, and the top-10 rate for `DeepDebug` (traces) is 97%. Note that these are highly nontrivial bugs, with a median length of two lines that must be edited to produce a fix. See table 5.

Table 5: Patch success results on our benchmark of neural bugs with executable tests.

|  | DeepDebug (backtrans) | DeepDebug (traces) |
| --- | --- | --- |
| Top-1 Success Rate | 357/523 (68%) | 393/523 (75%) |
| Top-10 Success Rate | 472/523 (90%) | 509/523 (97%) |

## 6 FUTURE WORK

Having code with executable tests opens up a world of opportunities for algorithmic debugging with machine learning. We are interested in genuinely iterative fixing (in contrast to independently generating edits), moving beyond the assumption that the bug has already been localized to a single method, and expanding to other popular languages. Most importantly we are interested in deploying a command-line interface to our tool.

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
