# OpenReview forum: "DeepDebug: Fixing Python Bugs Using Stack Traces, Backtranslation, and Code Skeletons"
_ICLR.cc/2022/Conference — ICLR 2022 Submitted_

### Official Review · Reviewer_sVvX · 2021-10-31

**Correctness:** 3
**Technical Novelty And Significance:** 2
**Empirical Novelty And Significance:** 3
**Recommendation:** 5
**Confidence:** 3

**Details Of Ethics Concerns:**

No significant ethical concerns.

**Main Review:**

1. This manuscript involves a complex set of 4-5 dataset for training and evaluation. The authors did a good job in presenting the datasets and their relations in Figure 1. But I still find a number of holes in the description of the datasets.
- Section 4.2: How are the 100,000 Python repositories selected from GitHub? Is there any additional filtering beyond minimum # of stats (10)? It seems that on GitHub there should be more than 100,000 Python repositories that meet this criterion. Is the number 100,000 just a round number or is it the exact number? Please provide numbers in as precisely as possible. Is this dataset available for examination by the public or by reviewers anywhere? I would request the dataset to be made available to make it easier to review. I understand the size of the dataset will be large, but it can be open sourced as a list of URLs or pointers to original datasets on GitHub.
- Section 4.2: The authors did not report the # of commits used to train the forward and backward edit models. Like I asked above, are these commit examples available for examination anywhere? Was the human-written commit messages used in anyway? How were the commits split into the train / dev / test subset? During the training of the forward and backward edit models, what hyperparameters were tuned and how? What if a commit touches multiple Python methods and/or multiple Python files? Were the commit treated as a single example or multiple examples? Without these details, it is hard for an independent research or lab to assess or reproduce the results.
- Section 4.2: The authors mentioned in the first paragraph of Sect. 4.2 that the data is "extremely noisy". But they did not talk about the implication f the noisiness or whether they did anything to mitigate its effect.
- Section 4.3: The authors mention that the model(s) generate syntax errors, often due to premature EOS tokens. But the authors didn't say how they dealt with these outputs with syntax errors. Did they simply discard them from the evaluation? Were they replaced with additional outputs by the model until a desired number of synthetic bugs without syntax error is reached?
- Section 4.4.2: Of the 10K repositories, how many individual tests were used? How were the tests split into train / dev / test subsets? Like asked above, is the container (Docker container?) and the tests available in open source for examination and reproduction?
- Section 4.4.3: How many such synthetic bugs with tests were there?
- Section 4.5: It seems that including all this additional context can lead to very long inputs, which may not fit into the 1024-token window. What did the author do to handle such out-of-bound situations? Were different context lines ranked by priorities and only the highest-priority ones preserved during the truncation? If so, the authors should give more details for the ranking algorithm. They should also report statistics on the number of different kinds of context lines that were included. Or even better, the code skeleton dataset should be made available for examination and reproduction.
- Section 4.5: When obtaining the code lines from methods other than the focal method, did the authors use a recursive approach? For example if the focal method uses another method, which in turn uses a third method, were both methods included in the skeleton? Could the method be from a different file in the same GitHub repository? From a PyPI package installed via setup.py? From a built-in Python module? These are important details without which it'd be hard to assess or reproduce the experiments.

2. The concept of "false positive" is mentioned in Section 2.2 and Table 3. However, to me the author didn't define false positives clearly. Does "false positive" mean fixes that are only test suite adequate, but not genuine fixes? In the context of the QuixBugs benchmark, how did the authors determine whether a fix is a false positive?

3. Questions about Table 3:
- Why not report results from DeepDebug trained without backtranlation augmentation as an ablation experiment? This would help quantify the benefit of backtranslation.
- To quantify the hit rate of the DeepDebug model, please report the # of bugs fixes and other metrics with fewer than 100 random samples (e.g., 10, 50). One minute is still very long for online use cases (e.g., imagine this being integrated into a code editor or IDE). It would be informative to report accuracy numbers under smaller # of attempts and shorter latency budgets.

4. Section 5.1: 2nd sentence of the 2nd paragraph: "Surprisingly, the backtranslation model actually performs worse on reversed
commit data, which is consistent with its bias towards constructively fixing buggy functions." seems written incorrectly.

5. General questions for discussion: The QuixBugs benchmark that the DeepDebug model of this paper is evaluated on contains mostly self-contained programs that do not interact with other modules or libraries in complex ways. This significantly reduces the difficulty of bug fixing. Real-life programs generally are organized into multiple modules (code files). They interact not only among themselves but also with other libraries (e.g., packages from PyPI) in complex ways. Some discussion should be added to describe how these challenges might or might not be solvable in the future in the DeepDebug approach.

**Summary Of The Paper:**

This manuscript describes DeepDebug a transformer-based mode that performs code repair on Python methods. Specifically, the model is pretrained on 200k Python code repositories and fine-tuned on a number of datasets consisting of bug fix commits and augmentation through synthetic bugs. The model takes Python methods with bugs as the input and outputs potentially bug free versions of the method. The main contribution of the paper, if held up through the questions I ask below, include:
- An innovative application of the backtranslation approach on the code repair domain.
- Setting a new state-of-the-art (SOTA) performance for code repair on the Python problems from the QuixBugs benchmark, beating previous SOTA (e.g., CoCoNut) not only in the percentage of bugs fixed, but also in using fewer attempts and significantly less latency budget.

While these contributions are exciting and potentially pushing the boundary of deep learning-based code repair, there are some major issues  with how the methodology is described, how the results are presented, and how the implications of the results are discussed. These critique points from me are listed in the "Main Review" section below.

**Summary Of The Review:**

In summary, this paper presents some potentially state-of-the-art performance on neural code repair. But there are a large number of nontrivial details that are left out from the methods, results and discussion parts of the manuscript, which I respectfully request the authors to address. If the authors can satisfactorily address these questions and concerns, the manuscript will be a strong candidate for publication in this conference in my opinion.

---

> ### Author Response · Authors · 2021-11-09
> **author's response**
>
>  >>> Section 4.2: How are the 100,000 Python repositories selected from GitHub?
>
> They were public python repos with at least ten stars that were edited sometime between 2015 and 2020. You can look at examples by using GitHub's advanced search option.
>
>  >>> Section 4.2: (Several questions)
>
> We didn't do any hyperparameter tuning. We used hyperparameters similar to BART. We broke up each commit into smaller edits that just edit individual functions, leaving 1.1 million functions. We only used commit messages for filtering. You can see examples of commit messages by cloning some python repos and looking at their commit histories.
>
> >>> Section 4.2: The authors mentioned in the first paragraph of Sect. 4.2 that the data is "extremely noisy". But they did not talk about the implication f the noisiness or whether they did anything to mitigate its effect.
>
> We did some filtering based on the commit message (if it includes the word "fix"). Ultimately we resorted to backtranslation, the main focus of the paper.
>
> >>> Section 4.3: The authors mention that the model(s) generate syntax errors, often due to premature EOS tokens.
>
> We count those completions as failures.
>
> >>> Section 4.4.2: Of the 10K repositories, how many individual tests were used? How were the tests split into train / dev / test subsets? Like asked above, is the container (Docker container?) and the tests available in open source for examination and reproduction?
> >>> Section 4.4.3: How many such synthetic bugs with tests were there?
>
> There were around 500k-1m tests and around as many functions covered by those tests. We produced 5 synthetic bugs for each of those functions. We split at the repo level to form the train/dev/test sets. We never got around to open-sourcing, sorry.
>
> >>> Section 4.5: It seems that including all this additional context can lead to very long inputs, which may not fit into the 1024-token window. What did the author do to handle such out-of-bound situations? Were different context lines ranked by priorities and only the highest-priority ones preserved during the truncation?
>
> Yes, that's exactly right. The order of priorities is: focal buggy function > function's class definition > imports > function signatures> function docstrings > function bodies (in the last three cases preferring functions in the same class)
>
>  >>> Section 4.5: When obtaining the code lines from methods other than the focal method, did the authors use a recursive approach? For example if the focal method uses another method, which in turn uses a third method, were both methods included in the skeleton?
>
> We limited to within the focal file.
>
>  >>> The concept of "false positive" is mentioned in Section 2.2 and Table 3. However, to me the author didn't define false positives clearly. Does "false positive" mean fixes that are only test suite adequate, but not genuine fixes? In the context of the QuixBugs benchmark, how did the authors determine whether a fix is a false positive?
>
> Yes, that's the right definition. We reported that we had one false positive on QuixBugs.
>
>  >>> Why not report results from DeepDebug trained without backtranslation augmentation as an ablation experiment? This would help quantify the benefit of backtranslation.
>
> In our opinion, the main advantage of backtranslation is that it enables training on traces. I'd expect the gains from pure backtranslation training without traces to be relatively small. Note that we did report a cross-entropy comparison, where backtranslation was giving a 10% boost or so. We omitted the ablation you proposed, where again there was a small improvement (less than 10%).
>
> >>> To quantify the hit rate of the DeepDebug model, please report the # of bugs fixes and other metrics with fewer than 100 random samples (e.g., 10, 50). One minute is still very long for online use cases (e.g., imagine this being integrated into a code editor or IDE). It would be informative to report accuracy numbers under smaller # of attempts and shorter latency budgets.
>
> The number of fixes with ten attempts was close to one tenth the number of fixes with 100 attempts, (intuitively it would be somewhere between a tenth and half as many). I think it's a fine user experience if the tool can run in the background.
>
> >>> Section 5.1: 2nd sentence of the 2nd paragraph: "Surprisingly, the backtranslation model actually performs worse on reversed commit data, which is consistent with its bias towards constructively fixing buggy functions." seems written incorrectly.
>
> The backtranslation model was trained to fix bugs, so it's arguably a good thing if it's worse at writing backwards commits, which tend to be more destructive.
>
>  >>> General questions for discussion: The QuixBugs benchmark that the DeepDebug model of this paper is evaluated on contains mostly self-contained programs that do not interact with other modules or libraries in complex ways.
>
> Very true. Hopefully bugpatching models will be better able to tackle real-world problems soon.

---

### Official Review · Reviewer_vC4c · 2021-11-02

**Correctness:** 4
**Technical Novelty And Significance:** 2
**Empirical Novelty And Significance:** 2
**Recommendation:** 5
**Confidence:** 3

**Main Review:**

Strengths:
- Authors trained a model on reversed commit data to generate synthetic bugs.
- Authors used the bug generation model to generate buggy code for training bug fixing model.
- Authors also used generated buggy code to produce stack traces with errors. These stack traces were then used to train the bug fix model.
- Authors used additional context such as import statements, class definitions, docstrings, etc. to further improve the bug fix model.
- Authors show a significant improvement on QuixBugs benchmark both in total fixes found and decrease of false positives.

Weaknesses:
- Fine tuning pre-trained models for bug finding and fixing is not novel.
- Using reversed commit data to generate bugs is not novel. It is possible that the authors' approach for such bug generation is better than what others have done, but this is not demonstrated.
- It is possible that using DeepDev-py as an intermediate model and then using two phase training without and with traces yields superior results. It is however not evaluated which parts of this process are important/significant and which are not.
- Limited comparison to other bug fixing tools.

If paper is accepted or in any other future versions, it would be good if authors compared their work with
"Self-Supervised Bug Detection and Repair", M. Allamanis, H. Jackson-Flux, M. Brockschmidt. NeurIPS 2021

Minor comments:
- Page 1 Abstract - Possibly change "timeout" to "execution time"?
- Page 4 - Replacing literals with placeholders might be removing fixes/bugs. Is this an issue? Please justify/comment.
- Page 6: Figure 4: is capital S in Sax intentional?


Not a critique to authors, but I am looking forward to some standardization of the bug detection/fixing benchmarks. Different authors/groups use different test sets/benchmarks making it hard to compare respective papers.


**Summary Of The Paper:**

Authors propose couple approaches to train models for automatically debugging Python programs.
Authors create bug generation model that generates training set with automatically added bugs. The bugs are used in code repositories with tests to obtain stack traces of errors. These stack traces are used to train debugging models that include stack traces. Authors also propose including context such as import statements, class declarations, and docstrings to improve performance of the debugging model. Authors show significant improvement on QuixBugs benchmark.

**Summary Of The Review:**

Authors present a bug fixing approach using tuned transformer models. Authors generate synthetic bugs by using reversed commit data. They additionally train the model on stack traces with errors caused by synthetic bugs. Authors propose using additional textual context to improve bug fixing. All this results in improved bug fixing model. I am uncertain if the contribution is significant enough for accepting the paper.

---

> ### Author Response · Authors · 2021-11-09
> **author's reponse**
>
> As we say in another response:
>
> The paper's novelty is that we collect by far the largest and most diverse dataset of executable tests that cover known bugs (which we produce via backtranslation). We also introduce the notion of code skeletons, which are a sophisticated way to fit a file into a limited context window.
>
> We choose to focus on bugs that come with executable tests in our evals. Other benchmarks are much noisier. We focus on a broader class of bugs than Allamanis et al. 2021 (also that paper wasn't out when we completed this work).

---

> > ### Comment · Reviewer_vC4c · 2021-12-01
> > **Thank you for your response**
> >
> > Thanks for elaboration on the paper's contributions.

---

### Official Review · Reviewer_K9aQ · 2021-11-02

**Correctness:** 2
**Technical Novelty And Significance:** 2
**Empirical Novelty And Significance:** 2
**Recommendation:** 3
**Confidence:** 4

**Main Review:**


Strengths
=========
1) Extensive Data used in training and learning the buggy function. The neural edit model is quite a refreshing idea.
2) Good Result on QuixBug and outperform baseline by significant results
3) Interesting idea of using the skeleton around the function. Employ information on the program (e.g. skeleton) in training and generating the patches.


Weakness
========
1) I would argue that the novelty in the paper is quite lacking. The model that the authors employed is just a pre-trained model. Although the data collection are extensive, but it does not really show any novelty in patch generation.

2) I have some questions about using backtranslation on evaluating the model. I am quite confused about the forward edit and backward edit. I don't think it is defined anywhere on the paper.
Furthermore, what is the point of using cross-entropy in evaluating the model? I do understand it shows that the model learns better using different edits and contexts, but it does not really reflect that it fixes the bugs. It will be better if you evaluate the model on the quix and report their results.   What do you use cross-entropy in evaluating the model?



**Summary Of The Paper:**

This paper proposes an approach to fixing synthetic python bugs by using the backtranslation model. The idea is to employ four different
types of datasets and employ BART to learn from them. They employ backtranslation in evaluating the trained model and outperform the previous baseline of over 50%. They further evaluate their model with test cases and achieve a fixed rate of over 90%

**Summary Of The Review:**

Overall, it is my opinion that the data collection are extensive and covers many different scenarios, although it is quite confusing to read it due to many terms and different type of bugs.
However, the novelty in the learning-based auto program repair is quite lacking. Furthermore, the evaluation is not suitable (e.g., uses cross-entropy to evaluate the model,  where you can actually report the difference model performance on quixbug). The paper will be much comprehensible if you employ similar metrics for all experiments and evaluations (e.g., top-1 fixes)

---

> ### Author Response · Authors · 2021-11-09
> **author's reponse**
>
> >>> I would argue that the novelty in the paper is quite lacking. The model that the authors employed is just a pre-trained model. Although the data collection are extensive, but it does not really show any novelty in patch generation.
>
> Our approach is novel in three different ways, as stated in the title. We collect by far the largest and most diverse dataset of executable tests that cover known bugs (which we produce via backtranslation). We also introduce the notion of code skeletons, which are a sophisticated way to fit a file into a limited context window.
>
>
> >>> I have some questions about using backtranslation on evaluating the model. I am quite confused about the forward edit and backward edit. I don't think it is defined anywhere on the paper.
>
> The forward edit model was trained on commit data as is standard (i.e. the source is the code before the commit and the target is the code after the commit). More precisely we break up commits into edits to individual functions, thus getting multiple src-tgt pairs per commit. To train the backwards edit model we simply reverse src and tgt, thus training the model to to backward commits. Sorry if this is unclear.
>
> >>> Furthermore, what is the point of using cross-entropy in evaluating the model? I do understand it shows that the model learns better using different edits and contexts, but it does not really reflect that it fixes the bugs. It will be better if you evaluate the model on the quix and report their results. What do you use cross-entropy in evaluating the model?
>
> Cross-entropy evals are easy to compute since they're given for free during training, and intuitively there should be a tight correlation between cross-entropy and actual performance. In our opinion, the main advantage of backtranslation is that it enables training on traces. I'd expect the gains from pure backtranslation training without traces to be relatively small. We omitted some experiments showing a small improvement (less than 10%, in line with the cross-entropy numbers).  I agree it would be good to have more evals on executable benchmarks and it might have been better to include all such evals rather than merely the ones we thought were most important.

---

### Official Review · Reviewer_Rbe2 · 2021-11-02

**Correctness:** 3
**Technical Novelty And Significance:** 2
**Empirical Novelty And Significance:** 2
**Recommendation:** 5
**Confidence:** 4

**Main Review:**

Pros:

+ The idea of data augmentation with back translation is interesting although not new.
+ Code skeleton to augment the code context.


Cons:
- QuixBugs benchmark is too small to evaluate the effectiveness of the model, while cross-entropy train losses are not useful in my opinion.
- No comparison to state-of-the art models of the same kind such as PL-BART, CodeT5, etc.
- Lack of discussion to recent work on self-supervised bug detection, e.g., Allamanis et al., 2021.

Questions:

1. Back translation is interesting but when the model is trained on commit data, the neural bug patterns will be at most the same as the ones seen in commit (?), so I don’t know how back translation would be helpful. At least, one experiment is to compare a model trained and evaluated on <before, after> code pair with the same model augmented by neural bugs.

2. Is there a reason why Python is the main focus? Recently, a lot of work (CodeBERT, PL-BART, CodeT5) were evaluated on the Patches of the Wild Java benchmark. Fine-tuning DeepDebug on that dataset seems straightforward to compare to those models.

For Python benchmarks, maybe PyPI benchmark and the random dataset therein (Allamanis et al., 2021) can be used.

3. You state that “make use of the plenitude of relatively bug-free code on GitHub”. Do you have a rough estimate of the percentage of the Github code that is bug-free? Without good filtering heuristics (e.g., based on code review), I think that augmentation data would be very noisy as well.


**Summary Of The Paper:**

In the paper, the authors propose DeepDebug, a Transformer-based approach for automated program debugging. DeepDebug is equipped with multiple sources of data such as stack traces and code skeleton, pre-trained and augmented with synthetic bugs.

**Summary Of The Review:**

See above.

---

> ### Author Response · Authors · 2021-11-09
> **responding to all points**
>
> Responding point by point.
>
>
> >>> QuixBugs benchmark is too small to evaluate the effectiveness of the model, while cross-entropy train losses are not useful in my opinion.
>
> The improvements we get on QuixBugs are gigantic, which means they can't possibly just be noise. Cross-entropy losses are always useful, as a complement to running tests if nothing else.
>
> >>> No comparison to state-of-the art models of the same kind such as PL-BART, CodeT5, etc.
>     Lack of discussion to recent work on self-supervised bug detection, e.g., Allamanis et al., 2021.
>
> You pointed out in your previous criticism that metrics like cross-entropy loss aren't as useful, which is why we choose to focus on bugs that come with executable tests in our evals. We focus on a broader class of bugs than Allamanis et al. 2021 (also that paper wasn't out when we completed this work).
>
>
> Questions:
>
>  >>> Back translation is interesting but when the model is trained on commit data, the neural bug patterns will be at most the same as the ones seen in commit (?), so I don’t know how back translation would be helpful. At least, one experiment is to compare a model trained and evaluated on <before, after> code pair with the same model augmented by neural bugs.
>
> Backtranslation is essential for getting localized bugs that we can be confident are bugs. It's necessary for collecting stack trace data at scale.
>
> >>> Is there a reason why Python is the main focus? Recently, a lot of work (CodeBERT, PL-BART, CodeT5) were evaluated on the Patches of the Wild Java benchmark. Fine-tuning DeepDebug on that dataset seems straightforward to compare to those models.
>
> Patches in the Wild doesn't have executable tests. Personally, I am much more familiar with Python than Java and can better evaluate a Python model.
>
>  >>> You state that “make use of the plenitude of relatively bug-free code on GitHub”. Do you have a rough estimate of the percentage of the Github code that is bug-free? Without good filtering heuristics (e.g., based on code review), I think that augmentation data would be very noisy as well.
>
> I don't have numbers, but intuitively a large majority of the code on github does what it's supposed to do.

---

> > ### Comment · Reviewer_Rbe2 · 2021-11-19
> > **Thank you for the response**
> >
> > > Allamanis et al. 2021 wasn't out when we completed this work.
> >
> > Just to clarify, that paper has been online since May, 2021, so I wasn't trying to ask for something unreasonable. Since your submission is closely related to that work, I believe it is appropriate to discuss and compare where applicable.
> >
> > Overall, I think the paper has several interesting ideas, but it perhaps needs more work, which I mentioned in my review, before it is ready for acceptance. I will keep the same evaluation.

---

### Decision · Program_Chairs · 2022-01-20

**Decision:**

Reject

**Comment:**

Metareview:

This paper proposes a transformer-based automated program debugger, called DeepDebug. All reviewers agree that the addressed problem is interesting. However, there was a consensus among reviewers regarding concerns in terms of the novelty and the lack of comparisons with other.
In general, all reviewers consistently gave a score that is below the acceptance threshold. This paper is of interest to the ICLR audience, but current form is not ready for acceptance.

Summary Of Reasons To Publish:

-Good Result on QuixBug

-Synthetic bug generation

Summary Of Suggested Revisions

-Comparisons in/with other datasets/tools